# Observation of exceptional points in reconfigurable non-Hermitian vector-field holographic lattices

Choloong Hahn[1], Youngsun Choi[1], Jae Woong Yoon[1], Seok Ho Song[1], Cha Hwan Oh[1] & Pierre Berini[2,3,4]

Recently, synthetic optical materials represented via non-Hermitian Hamiltonians have attracted significant attention because of their nonorthogonal eigensystems, enabling unidirectionality, nonreciprocity and unconventional beam dynamics. Such systems demand carefully configured complex optical potentials to create skewed vector spaces with a desired metric distortion. In this paper, we report optically generated non-Hermitian photonic lattices with versatile control of real and imaginary sub-lattices. In the proposed method, such lattices are generated by vector-field holographic interference of two elliptically polarized pump beams on azobenzene-doped polymer thin films. We experimentally observe violation of Friedel's law of diffraction, indicating the onset of complex lattice formation. We further create an exact parity-time symmetric lattice to demonstrate totally asymmetric diffraction at the spontaneous symmetry-breaking threshold, referred to as an exceptional point. On this basis, we provide the experimental demonstration of reconfigurable non-Hermitian photonic lattices in the optical domain and observe the purest exceptional point ever reported to date.

[1] Department of Physics, Hanyang University, Seoul 133-791, Korea. [2] School of Electrical Engineering and Computer Science, University of Ottawa, 800 King Edward Avenue, Ottawa, Ontario, Canada K1N 6N5. [3] Department of Physics, University of Ottawa, 150 Louis Pasteur, Ottawa, Ontario, Canada K1N 6N5. [4] Centre for Research in Photonics, University of Ottawa, 25 Templeton Street, Ottawa, Ontario, Canada K1N 6N5. Correspondence and requests for materials should be addressed to J.W.Y. (email: jaeong.yoon@gmail.com) or to S.H.S. (email: shsong@hanyang.ac.kr).

E ngineered optical materials have been extensively studied in the pursuit of new materials with exotic properties unavailable from natural substances, especially within the context of photonic crystals[1] and metamaterials[2]. In these artificial systems, new functionalities have been found, generally by manipulating the real part of permittivity (or permeability) to achieve a desired spatial distribution. Synchronizing such systems with carefully configured imaginary permittivities, a new approach inspired by an abstract theoretical concept based on non-Hermitian Hamiltonians, is now of great interest in both theory and experiment. This emerging strategy is motivated in part by the experimental feasibility of optical systems for mimicking parity-time (PT) symmetric quantum systems without any conflict with the Dirac-von Neumann's Hermiticity axiom for physical observables. In seminal papers on PT-symmetric Hamiltonians[3–5], Bender and coworkers showed that such Hamiltonians display completely real-valued energy spectra below certain phase-transition thresholds, referred to as exceptional points. This property leads to skewed eigenvector spaces where the system's time evolution and spectral features are remarkably distinct from standard Hermitian systems[6–9].

In optics, one can produce a system having a PT-symmetric Hamiltonian by engineering a complex dielectric distribution $\varepsilon(\mathbf{r})$ that is invariant under spatial inversion and has simultaneous loss–gain interchange, that is, $\varepsilon^{\star}(-\mathbf{r}) = \varepsilon(\mathbf{r})$, where $\mathbf{r}$ is a position vector. Therefore, the notion of PT symmetry and its associated phenomena in optics are accessible experimentally, revealing abstract non-Hermitian dynamics and further providing technologically relevant underlying physics. Introducing several representative reports, among a wide variety of theoretical and experimental works along this line, Regensburger et al.[10] proposed a coupled fibre-optic network structure for PT-symmetric time-domain lattices as an experimental testbed. They experimentally showed intriguing optical effects such as exact symmetry breaking of the system's eigenmodes at the exceptional point, Bloch oscillations and nonreciprocal reflection, which are characteristic features of skewed eigenvector spaces. Moreover, purely spatial, complex photonic lattices have shown unconventional beam dynamics associated with spectrally singular Bragg scattering[6], asymmetric or solitary optical propagation[11–14] and counter-intuitive uniform-intensity wave solutions in non-uniform media[15]. Exploiting these phenomena for device applications, nonreciprocal transmission[16] and unidirectional-reflection elements[17–19] based on photonic integrated circuits were experimentally demonstrated without introducing any nonlinear or gyrotropic material to break time-reversal symmetry. Of fundamental importance in this context is the development of versatile platforms where arbitrary complex optical potentials can be precisely configured and non-Hermitian optical effects experimentally explored.

Although some proof-of-concept experiments have been reported, synthesizing non-Hermitian optical structures generally demands challenging high-precision fabrication involving multiple etching and deposition steps with deep-subwavelength inter-step alignment tolerance[16,17,19–22]. For example, a PT-symmetric dielectric function demands $\mathrm{Re}(\varepsilon)$ and $\mathrm{Im}(\varepsilon)$ profiles of opposite spatial parity. Creating such a profile is clearly nontrivial in the context of conventional nanophotonic architectures. Consequently, further realizing such structures in reconfigurable platforms is presently a formidable task in the optical domain. Taking a completely different approach in this paper, we report comprehensive non-Hermitian photonic lattice generation using vector-holographic interference in azo-dye-doped polymer (azo-polymer) thin films. Under the influence of polarized periodic optical fields, azo dyes in polymer matrices simultaneously induce surface relief, birefringence and

dichroic sub-gratings due to molecular migration and reorientation triggered by photoisomerization and relaxation processes[23,24]. These sub-grating components induced by a single holographic vector field are precisely deployed to synthesize a desired non-Hermitian system. Therefore, robust and reconfigurable non-Hermitian photonic lattices can be generated using the proposed method.

## Results

**Formation of complex photonic lattices.** The essence of our proposed method is illustrated schematically in Fig. 1a. Two coherent electric fields $\mathbf{E}_1$ and $\mathbf{E}_2$ of different polarization states are incident to form a periodic pump field $\mathbf{E}_{\mathrm{P}}(x) \equiv \mathbf{E}_1 + \mathbf{E}_2 = \mathbf{E}_{\mathrm{P}}(x \pm \Lambda)$ in an azo-polymer thin film. $\mathbf{E}_{\mathrm{P}}$ leads to the simultaneous formation of a surface-relief grating and of a dichroic absorption grating, each contributing a modulation to the real and imaginary dielectric function, respectively. These two sub-grating components originate from distinct mechanisms: the former is generated by optical gradient forces and consequent migration of azobenzene–polymer complexes[22], while the latter is formed by polarization-induced reorientation of azo-dye molecules[23]. Considering these two mechanisms for a given periodic pump field $\mathbf{E}_{\mathrm{P}}$, the dielectric function $\varepsilon(x)$ in the azo-polymer film is written $\varepsilon(x) = \varepsilon_{\mathrm{avg}} + \Delta\varepsilon_{\mathrm{R}}(x) + i\Delta\varepsilon_{\mathrm{I}}(x)$ with the real ($\Delta\varepsilon_{\mathrm{R}}$) and imaginary ($\Delta\varepsilon_{\mathrm{I}}$) modulations determined by

$$\Delta\varepsilon_{\mathrm{R}}(x) = C_{\mathrm{R}}\chi \frac{\partial}{\partial x} \langle \mathbf{E}_{\mathrm{P}} \cdot \nabla E_{\mathrm{P}x} \rangle_t, \quad (1)$$

$$\Delta\varepsilon_{\mathrm{I}}(x) = C_{\mathrm{I}} \frac{2}{1 + \sqrt{1 - \sin^2 2\alpha \cdot \sin^2 \varphi_{xy}}} \left( |E_{\mathrm{P}x}|^2 - |E_{\mathrm{P}y}|^2 \right). \quad (2)$$

Here, $\chi$ is the electric susceptibility of the host polymer, $\langle \cdots \rangle_t$ implies time-averaging of the argument, and $\alpha = \tan^{-1}(|E_{\mathrm{P}y}/E_{\mathrm{P}x}|)$ and $\varphi_{xy} = \arg(E_{\mathrm{P}x}) - \arg(E_{\mathrm{P}y})$ are local polarization parameters. The empirical constants $C_{\mathrm{R}}$ and $C_{\mathrm{I}}$ are fixed for a given dye doping concentration and film thickness. See Supplementary Note 1 and Supplementary Figs 1 and 2 for additional details on these relations. We note in equations (1) and (2) that the dominant contributions result from the second derivative $\partial^2 |E_{\mathrm{P}x}|^2/\partial x^2$ of the intensity for $\Delta\varepsilon_{\mathrm{R}}(x)$ and the polarization contrast $|E_{\mathrm{P}x}|^2 - |E_{\mathrm{P}y}|^2$ for $\Delta\varepsilon_{\mathrm{I}}(x)$. Therefore, one can judiciously control relative magnitudes and phases of $\Delta\varepsilon_{\mathrm{R}}(x)$ and $\Delta\varepsilon_{\mathrm{I}}(x)$ by tuning the polarization state of $\mathbf{E}_1$ and $\mathbf{E}_2$ to form a desired distribution in the single vector-holographic pump field $\mathbf{E}_{\mathrm{P}}$.

We confirm experimentally the proposed concept using 100-nm-thick PMMA thin films doped with Disperse Red 1 (Sigma-Aldrich) azo-dye at a 15% molecular concentration. A Nd:YAG laser operating at a wavelength of 532 nm and supplying a continuous-wave power of 30 mW is used as a coherent pump source. We show two representative combinations of $\mathbf{E}_1$ and $\mathbf{E}_2$ along with the consequent $\mathbf{E}_{\mathrm{P}}$, and dielectric function modulations $\Delta\varepsilon_{\mathrm{R}}(x)$ and $\Delta\varepsilon_{\mathrm{I}}(x)$ over one period (1.356 μm) in Fig. 1b,c. The experimental profiles measured by phase-shifting interferometry agree quantitatively with theoretical predictions using equations (1) and (2). See Supplementary Notes 2 and 3 for the description of the measurement method and Supplementary Figs 3 and 4 for details of the experimental configuration. The generated complex photonic lattices can be conveniently expressed by first-harmonic sinusoidal modulations in the real and imaginary dielectric functions:

$$\Delta\varepsilon(x) = \Delta\varepsilon_{\mathrm{R}}(x) + i\Delta\varepsilon_{\mathrm{I}}(x) = \Delta\varepsilon_0[(1 - \xi)\cos(Kx) + i\xi\cos(Kx - \delta)],$$

$$(3)$$

where $K = 2\pi/\Lambda$ with $\Lambda$ being the period of the modulation.

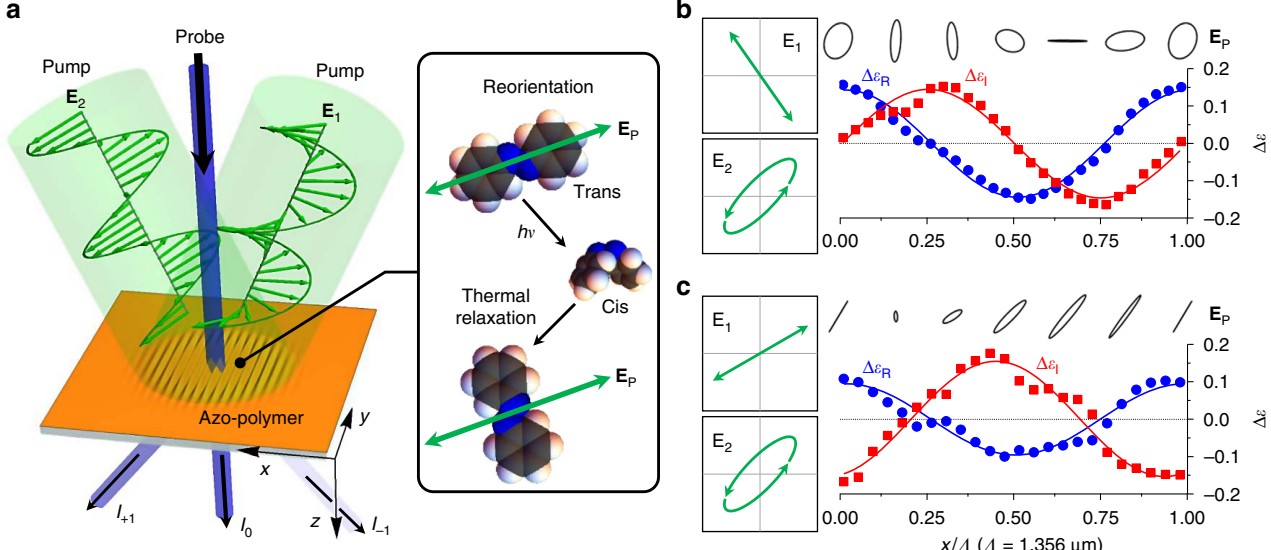

**Figure 1 | Complex photonic lattice generation with vector-holographic interference in azo-polymer thin films.** (**a**) Schematic illustration of pump-probe configuration on an azo-polymer thin film. Two coherent incident fields $\mathbf{E}_1$ and $\mathbf{E}_2$ generate a holographic pump field $\mathbf{E}_P$ that induces concurrent real and imaginary sub-gratings as results of photo-isomerization-induced molecular migration and reorientation. (**b,c**) Generated real $\Delta\varepsilon_R$ and imaginary $\Delta\varepsilon_I$ sub-grating profiles along the x axis for two representative combinations of $\mathbf{E}_1$ and $\mathbf{E}_2$ polarization states. The symbols correspond to the profiles measured by phase-shifting interferometry and the solid curves are theoretical (equations (1) and (2)). Case (**b**) satisfies precisely the condition for operation at the PT-symmetry breaking threshold, that is, $\Delta\varepsilon^{\star}(-x) = \Delta\varepsilon(x)$ with identical modulation depths in $\Delta\varepsilon_R$ and $\Delta\varepsilon_I$.

Here we define the balance factor $\xi \in [0, 1]$ and relative phase difference $\delta \in [-\pi, \pi]$ as the primary parameters determining the non-Hermitian properties of the lattice.

**Optical beam dynamics in the complex lattices.** The evolution of a photonic state through the lattice is described by:

$$|\mathbf{A}(z)\rangle = \exp(i\mathbf{H}z)|\mathbf{A}(0)\rangle. \quad (4)$$

Considering configurations allowing only the $p = -1, 0$ and $+1$ diffraction orders as propagating through the lattice, we express the state vector such that $|\mathbf{A}(z)\rangle = [A_{-1}(z)\ A_0(z)\ A_{+1}(z)]^T$ with $A_p(z)$ being the amplitude of the $p$-th diffraction order. For a $y$-polarized plane wave, the $3 \times 3$ Hamiltonian matrix $\mathbf{H}$ is given by

$$\mathbf{H} = \frac{k_0}{2\varepsilon_{avg}^{1/2}} \begin{pmatrix} -K^2/k_0^2 & \eta_{-1} & 0 \\ \eta_{+1} & 0 & \eta_{-1} \\ 0 & \eta_{+1} & -K^2/k_0^2 \end{pmatrix} \quad (5)$$

where the constants $\eta_{\pm 1} = \{1 + [i\exp(\mp i\delta) - 1]\xi\}\Delta\varepsilon_0/2$ denote the Fourier coefficients of $\Delta\varepsilon(x)$ at the $\pm 1$ harmonic orders, respectively, and $k_0$ is the vacuum wavenumber. (See Supplementary Note 2 for details.) In this matrix representation, the parity operation $\mathbf{P}$ implies a matrix transpose such that $\mathbf{PH} = \mathbf{H}^T$ while the time reversal operation $\mathbf{T}$ is defined as a complex-conjugate transpose such that $\mathbf{TH} = (\mathbf{H}^\star)^T$. Therefore, the PT operation in our case yields $\mathbf{PTH} = \mathbf{H}^\star$. Consequently, the given Hamiltonian $\mathbf{H}$ is PT symmetric for $\eta_{\pm 1}^\star = \eta_{\pm 1}$ at $\delta = \pi/2$ in this formulation.

Solving the eigenvalue problem $\mathbf{H}|\mathbf{u}_\nu\rangle = \alpha_\nu|\mathbf{u}_\nu\rangle$ yields a set of eigenvectors $\{|\mathbf{u}_\nu\rangle\}$ and corresponding eigenvalues $\{\alpha_\nu\}$ representing stationary Floquet–Bloch modes in the lattice and group-transport momenta, respectively. We define the skewness parameters of the vector space defined by $\{|\mathbf{u}_\nu\rangle\}$ as $c_{\nu\mu} = |\langle\mathbf{u}_\nu|\mathbf{u}_\mu\rangle|$, producing values between 0 for orthogonal eigenvectors and 1 for eigenvectors merging at an exceptional point (EP). In Supplementary Note 2 and Supplementary Table 1, we provide

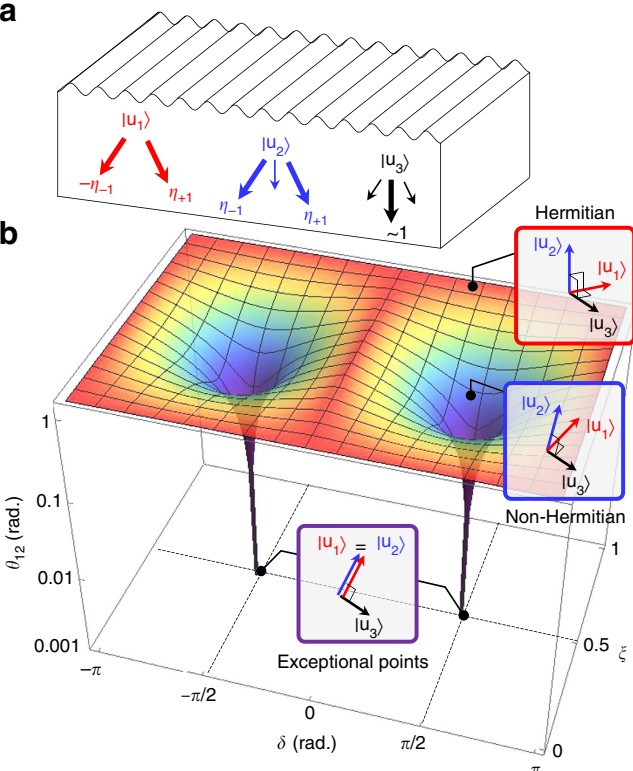

**Figure 2 | Skewed eigenvector structure.** (**a**) Three eigenvectors $|\mathbf{u}_1\rangle$, $|\mathbf{u}_2\rangle$ and $|\mathbf{u}_3\rangle$ and their relative amplitudes for the $p = -1, 0$ and $+1$ diffraction orders. (**b**) Cosine angle $\theta_{12} = \cos^{-1}(c_{12})$ of the skewness parameter as a function of the balance factor $\xi$ and phase difference $\delta$. The three insets are illustrative representations of the skewed vector space formed by $|\mathbf{u}_1\rangle$, $|\mathbf{u}_2\rangle$ and $|\mathbf{u}_3\rangle$.

closed-form expressions for $\alpha_\nu$, $|\mathbf{u}_\nu\rangle$, and $c_{\nu\mu}$. The character of the eigenvectors for the case of $\Delta\varepsilon(x)$ given by equation (3) is shown in Fig. 2a, which identifies the dominant amplitudes at the three allowed diffraction channels. The first eigenstate $|\mathbf{u}_1\rangle$ is an anti-symmetric combination of the $p = +1$ and $-1$ diffraction orders while $|\mathbf{u}_2\rangle$ is a symmetric combination of the $p = +1$ and $-1$ diffraction orders. These two eigenvectors form a merging pair at the EPs $(\xi, \delta) = (1/2, \pm\pi/2)$ and thus generate a skewed vector space. This property is clearly visualized in Fig. 2b where we present the cosine angle $\theta_{12} \equiv \cos^{-1}(c_{12})$ between $|\mathbf{u}_1\rangle$ and $|\mathbf{u}_2\rangle$ as a measure of the geometrical distance between these two states in the canonical Hilbert space. Intriguing non-Hermitian optical effects occur near the EPs characterized by $\theta_{12} = 0$. At these points, $|\mathbf{u}_1\rangle$ and $|\mathbf{u}_2\rangle$ are identical, so the vector space displays extreme skewness leading to interesting properties such as unidirectional or nonreciprocal energy transport[16–19,21,25] and a wormhole-like effect on state evolution in the Hilbert space[7–9].

To experimentally confirm the skewed subspace formed by $|\mathbf{u}_1\rangle$ and $|\mathbf{u}_2\rangle$, we observe violation of Friedel's law of diffraction[6,26,27] and totally asymmetric diffraction at exact EPs as immediate consequences of non-orthogonal eigenvectors. In detail, the contrast ratio $\Gamma \equiv I_{+1}/I_{-1}$ for the two first-order diffraction intensities produced under incidence by a single $y$-polarized plane wave as an optical probe, is directly connected to the skewness parameter $c_{12}$:

$$\Gamma = \frac{1 + c_{12}}{1 - c_{12}}, \qquad (6)$$

for $0 \le \delta < \pi$. For $-\pi < \delta < 0$, the same relation applies for the inverse ratio $I_{-1}/I_{+1}$ (see Supplementary Note 3 for the derivation). Clearly, $\Gamma = 1$ for Hermitian configurations having $c_{12} = 0$, that is, for $\theta_{12} = \pi/2$. As EPs are approached, that is, $(\xi, \delta) \rightarrow (1/2, \pm\pi/2)$, then $c_{12} \rightarrow 1$ and consequently $\Gamma$ diverges, implying that first-order diffraction becomes totally asymmetric. Although details are not presented in this letter, the EPs for a $x$-polarized probe are identical to those for a $y$-polarized probe and the same consequence in the contrast ratio applies once one switches $\delta$ to $\delta + \pi$ because the imaginary sub-grating profiles $\Delta\varepsilon_I(x)$ for orthogonal probe polarizations take opposite signs. Interestingly, this implies that polarization flipping between the $x$- and $y$-polarizations can induce a high-extinction switching of $\Gamma$ between 0 and $\infty$. In addition, a continuous change in $\Gamma$ is obtainable by rotating the probe polarization angle with respect to the periodicity axis of the lattice.

***In-situ* measurement**. An experimental set-up for real-time contrast-ratio measurements of complex lattice formations is

shown in Fig. 3. Two coherent pump beams from one Nd:YAG laser (operating at a wavelength of 532 nm and an output power of 30 mW) are prepared to form $\mathbf{E}_1$ and $\mathbf{E}_2$ with the desired polarization states using quarter-waveplate half-waveplate pairs to generate a vector-holographic pump field $\mathbf{E}_P$ in the azo-polymer film. A $y$-polarized probe beam from an Ar$^+$ laser, operating at a wavelength of 488 nm and an output power of 3 mW, is incident on the azo-polymer film at a surface-normal angle. The probe laser wavelength is selected to match the absorption maximum of the azo-dye polymer. The $I_{+1}$ and $I_{-1}$ intensities diffracted from the probe beam are monitored to acquire $\Gamma$ in the time window over which the complex lattice is formed.

We perform contrast-ratio measurements in the time domain with different polarization settings as outlined in the left inset of Fig. 4a. The fixed polarization parameters are the long-axis angle $\psi_2$ of $\mathbf{E}_2$ and the ellipticity $e_1$ and $e_2$ of $\mathbf{E}_1$ and $\mathbf{E}_2$, respectively, while the long-axis angle $\psi_1$ of $\mathbf{E}_1$ is varied from $-54.8°$ to $-40°$. From the results shown in Fig. 4a, we clearly observe maxima over the recording time that ranges from 6.5 to 8.5 min (highlighted as the orange band) for several cases of $\psi_1$. The time origin corresponds to the time at which the pump beam was turned on. The time response of the measurement is understood from the different timescales required for the formation of the real and imaginary sub-gratings. The imaginary sub-grating is formed from molecular reorientation processes which take a few seconds to build-up, while the real sub-grating forms from the comparatively slower migration of azobenzene–polymer complexes which needs $\sim 10^3$ s for complete build-up. Assuming a typical exponential relaxation for this process, the balance factor $\xi$ over the recording timespan $t$ follows:

$$\xi(t) = \frac{\xi_\infty}{1 - (1 - \xi_\infty)\exp(-t/\tau_R)}, \qquad (7)$$

where $\xi_\infty = \xi(t \rightarrow \infty)$ and $\tau_R$ is relaxation time of the molecular migration process for a given film thickness and pump-field intensity. Clearly, $\xi$ monotonically decreases from 1 at $t = 0$ to $\xi_\infty$ at $t \gg \tau_R$. Using phase-shifting interferometry, we estimate the time dependence of $\xi$ for $\psi_1 = -54.0°$ as plotted along the top of Fig. 4a with schematic illustrations of the corresponding $\Delta\varepsilon_R(x)$ and $\Delta\varepsilon_I(x)$ profiles. Thus, Fig. 4a clearly confirms that the contrast ratio $\Gamma$ is maximized for $\xi = 1/2$, where the eigenvector skewness parameter $c_{12}$ is highest for a given phase-difference parameter $\delta$. Moreover, the contrast ratio near $\xi = 1/2$ displays a diverging behaviour for $\psi_1 = -54.0°$ as this condition further satisfies the phase-difference requirement of $\delta = \pi/2$ for the EPs, as indicated in Fig. 4b which plots $\delta$ calculated as a function of $\psi_1$ using equations (1)–(3). In Fig. 4c, we compare the measured $\Gamma$ versus $\psi_1$ for $\xi = 1/2$ with the model based on equation (1)–(4). The quantitative agreement between the experiment and model again confirms that our method is efficient for accessing exact EPs associated with PT-symmetry breaking. We note that the experimental $\Gamma$ value, well in excess of 200, is remarkably higher than the previously reported values of $\sim 5.4$ in [19] and $\sim 14$ in [21], where the complex lattices were generated by lithographic and angle-deposition methods, respectively. It is also higher than the reported value of $\Gamma \sim 7 \pm 1$ generated by a complex photonic lattice in an azo-polymer film by Birabassov *et al.*[27] Their approach is based on spatially modulated spectral hole burning that permanently changes the material properties in an irreversible manner. To the best of our knowledge, the observed EP is purest among those ever obtained previously. Therefore, our proposal is very promising for generating precise non-Hermitian photonic lattices solely using finely controlled optical instruments.

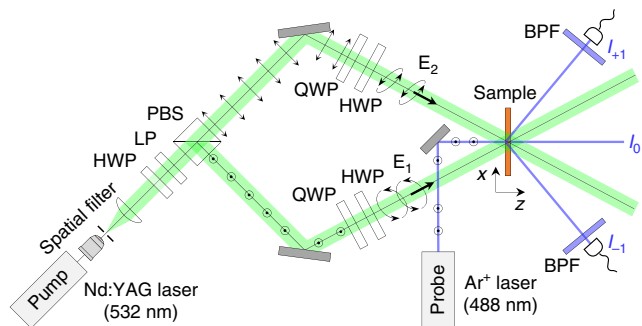

**Figure 3 | Real-time contrast-ratio measurement with complex lattice formation.** Abbreviated component labels denote half-wave plate (HWP), linear polarizer (LP), quarter-wave plate (QWP), polarizing beam splitter (PBS) and band-pass filter (BPF).

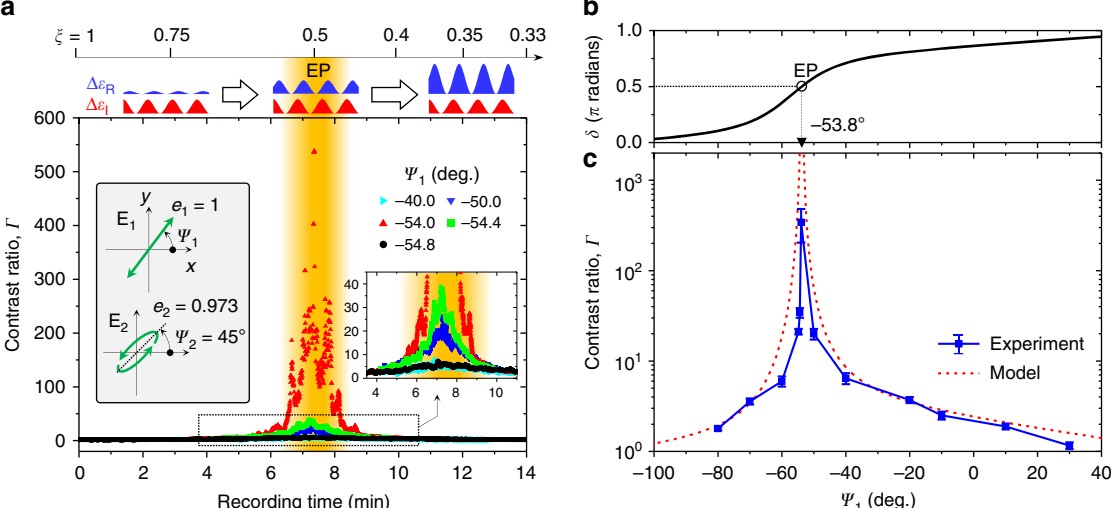

**Figure 4 | Radical violation of Friedel's law of diffraction near the PT-symmetry breaking threshold.** (**a**) Real-time measurement of the contrast ratio $\Gamma = I_{+1}/I_{-1}$ during complex photonic lattice formation for different polarization angles $\psi_1$ of $\mathbf{E}_1$. (**b**) Calculated dependence of the phase factor $\delta$ on the polarization angle $\psi_1$ of $\mathbf{E}_1$ using equations (1)–(3). (**c**) Contrast ratio $\Gamma = I_{+1}/I_{-1}$ as a function of $\psi_1$ with the other polarization settings fixed. The error bars correspond to the s.d. due to the dark current of the photodetector, which impacts $\Gamma$ for small $I_{-1}$ levels. The other polarization settings are indicated in the left inset to **a** and apply to **a**–**c**. In this inset, $e_{1,2}$ denote the polarization ellipticity of the pump fields $\mathbf{E}_{1,2}$.

## Discussion

Although the results summarized in Fig. 4 are taken from temporally varying complex lattices, stationary lattices with $\xi = \xi_\infty$ are obtained when the pump fields are continuously applied over a recording time $t \gg \tau_R$, which is of the order of 10 min in our experimental configuration. Here we have selected a sufficiently low $\xi_\infty$ ($\approx 0.28$) to produce a temporally tuned parametric scan over a sufficiently broad parameter range to clearly observe exceptional point behaviour. The $\xi_\infty$ value is in principle adjustable to higher values (up to 0.5) by tuning the pump-polarization parameters. Alternatively, incorporating a dielectric layer on top of an azo-polymer film provides another way to tune the $\xi_\infty$ value, following the elastic modulus and thickness of the dielectric layer. Turning off the pump fields leads to complete annihilation of the imaginary sub-lattice in a few seconds due to thermal relaxation at room temperature as the photo-isomerization and consequent molecular reorientation processes cease to occur. In contrast, the generated real sub-lattice persists in the form of a surface-relief grating until another pump or erasing fields drive new periodic gradient forces or planarizing potentials, respectively.

Importantly, the obtained results demonstrate the successful experimental realization of reconfigurable non-Hermitian photonic lattices in the optical domain. We note that reconfigurable non-Hermitian systems are of great importance for exploring chiral EP dynamics[28,29] and the associated asymmetric state interchange between orthogonal eigenmodes[30,31]. Although effects associated with EPs are observable in temporally static, space-variant systems[31], experimental realization of reconfigurable non-Hermitian systems provides rigorous means to study true time-domain dynamics and tunable devices taking advantage of temporally varying system parameters. Previously, a mechanically tunable microwave cavity was introduced to show a quasi-adiabatic state flip during parametric encircling around an EP[29]. More recently, a reconfigurable exciton-polaritonic non-Hermitian microcavity was generated using a pump-field imaging configuration[32]. However, migration of these concepts into photonic systems in the optical domain is formidable. A mechanically tunable single

or dual mode optical cavity should involve intricate micro or nano electromechanical system architectures with deep subwavelength fabrication tolerance. The pump-field imaging method for exciton-polaritonic tunable cavities is also experimentally futile in photonic systems because of extremely weak nonlinear interaction between pure photons. In this context, the proposed non-Hermitian holographic lattice platform alleviates such difficulties and thereby can be used to further investigate the chiral EP dynamics with remarkably improved experimental feasibility and parametric precision.

Previously, holographic lattice generation in azo-polymers has been widely studied to realize optical memory devices permitting stable reconstruction of data with multiple read–write–erase cycles below a certain photo-bleaching threshold intensity[33]. In our case, we have confirmed that significant photo bleaching is observed at a pump intensity of $1.7\,\mathrm{W\,cm^{-2}}$ for a recording time of over 2 h. In Supplementary Note 4 and Supplementary Fig. 5, we provide an experimental result on the complex lattice reconfiguration that shows stable formation of reconfigured complex lattices on a single sample spot with multiple write–erase–rewrite cycles. We confirm a stable reconfiguration time of ~1 min for a pump intensity of $1.7\,\mathrm{W\,cm^{-2}}$ for a total recording time of 40 min and a robust contrast ratio tuning in response to the varying pump-polarization parameters. In the experiment that leads to the data presented in Fig. 4, we have used again a single sample spot for multiple write–erase–rewrite cycles at a lower pump intensity of $30\,\mathrm{mW\,cm^{-2}}$. In this case, no photo-bleaching effects were identified and reconfiguring a complex lattice into another form takes a fairly longer time—about 10 min, as required for the stable formation of the real sub-lattice. In fact, a much shorter reconfiguration time is possible with a slight modification of the sample geometry. We note that creation and annihilation of the imaginary sub-lattice takes a few seconds, following the relaxation time of azo-dyes into the stable trans-phase state aligned in a preferred orientation for a given pump-field condition. Therefore, if a real sub-lattice is predefined, for example using a standard lithographic technique or other available means, one can readily obtain stable reconfiguration within a few seconds using this strategy.

In conclusion, we propose vector-holographic generation of non-Hermitian photonic lattices in azo-polymer thin films. We demonstrate, in both theory and experiment, versatile control of skewed eigenvector spaces in complex photonic lattices synthesized using the proposed method. We observe an extremely high asymmetry in the diffraction intensities at exact EPs, clearly confirming precise complex lattice formations. Notably, the proposed method generates reconfigurable non-Hermitian photonic lattices on demand. Moreover, replacing the azo-polymer with photorefractive media may lead to more efficient and dynamic complex lattices with greater controllability by means of electro-optic interaction or nonlinear optical properties[34]. Our approach is of strong interest for further investigation. For instance, introducing optical gain by doping azo-polymer films with laser dyes not only compensates average losses but also may enable observation of much broader classes of non-Hermitian optical effects including saturated-gain-induced nonreciprocal light transmission[35,36], spectral singularities and associated unconventional laser oscillation effects[6], optical solitons[11,13] and uniform-intensity Bloch wave beams[15]. On this basis, we envisage further experimental work on non-Hermitian spectral band engineering with sub-wavelength periodicity, the optical realization of encircling EPs with geometric phase effects, and attendant device applications such as switchable unidirectional couplers.

**Data availability**. The data that support the findings of this study are available from the corresponding authors on request.

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

## Acknowledgements

This research was supported in part by the Basic Science Research Program (NRF-2015R1A2A2A01007553) and by the Global Frontier Program through the National Research Foundation (NRF) of Korea funded by the Ministry of Science, ICT & Future Planning (NRF-2014M3A6B3063708).

## Author contributions

C.H. provided the original idea. C.H., S.H.S and C.H.O. initiated the work. C.H. and Y.C. performed the experiments. C.H., Y.C. and J.W.Y. developed the theory and models. All authors discussed and interpreted the results. J.W.Y., C.H., S.H.S. and P.B. wrote the manuscript. C.H. and Y.C. contributed equally to the work.

## Additional information

**Competing financial interests:** The authors declare no competing financial interests.

