## [Peer Review File · Nature Communications]

Reviewers' comments:

Reviewer #1 (Remarks to the Author):

Hahn and co-workers report a nice approach to construct complex gratings using a clever vector holographic interference illumination pattern of an azobenzene-doped polymer thin film. The authors provide a firm theoretical description and their experiments clearly demonstrate strong asymmetric diffraction between the $+1/-1$ orders at the exceptional point. The basic approach is flexible and adds to a relatively small handful of experimental routes to realize complex optical potentials, therefore I believe it will be a valuable contribution to the non-Hermitian optics community. A few points that would be helpful to address:

1. There is an existing report by Birabassov et. al., [Opt. Lett. 24, 1669 (1999)] that also describes asymmetric $+1/-1$ order diffraction through construction of a complex grating in an organic thin film through interference pattern illumination. The approach is different than that employed here, however, I would have expected it to be cited, particularly since Birabassov also used the azo dye Disperse Red 1.
2. A discussion of what it takes to write a temporally-stable complex grating would be helpful. While not an issue for the present experiments, in general, having a system that is continually changing in the manner of Fig. 4a is likely to hamper broader experimental use of this approach for exploring non-Hermitian optical potentials as well as some device applications. Additionally, providing a sense of how long it takes for the gratings to dissipate once the write beams are turned off would be helpful (at a given sample temperature).
3. The authors discuss a number of possible extensions of this approach toward the end of the manuscript, but notably do not mention the possibility of incorporating gain. As gain is an important aspect for broader investigations and applications of PT symmetry, I would expect at least some mention of routes to accomplish this (e.g. addition of a laser dye into the polymer film) and whether there are any particular challenges envisioned.

RECOMMENDATION

Invite the authors to revise their manuscript to address specific concerns before a final decision is reached

COMMENTS TO THE AUTHOR(S)

The manuscript by Hahn et al. entitled "Observation of exceptional points in reconfigurable non-Hermitian vector-field holographic lattices" presents experimental results of an azobenzene-doped polymer thin film configured to exhibit a PT-symmetric Hamiltonian. The results show good agreement with numerical models (also presented) and the authors claim the first experimental demonstration of reconfigurable non-Hermitian photonic lattices in the optical domain and observe the purest exceptional point to date. The findings enable a versatile platform for PT-symmetry investigations and device studies.

The work is, in the opinion of this reviewer, quite nice although some points raised in the manuscript should be expanded upon and clarified. The following comments substantiate this reviewer's opinion that the manuscript should be revised before a decision is reached:

- General interest: The work deals with the investigation of a new platform for studies of optical PT-symmetry, a very 'hot' field of research, with many interesting phenomena and applications already demonstrated within the last decade. I believe this context is well-presented in the paper. While I believe the manuscript will be of interest to others in the field due to its main claim of a reconfigurable non-Hermitian photonic lattice (note that this platform's accessibility versus competing platforms should be further discussed), I feel this claim could be made more convincing. Only in the concluding paragraph is it referenced that holograms in azo-polymers are rewritable (Ref. 28) and if the reconfigurability is to be the main claim, it should be more apparent when multiple configurations using *the same* azo-polymer are used (aka this part should be rewritten). The repeatability and ability to show different non-Hermitian configurations with a single film must be demonstrated to, in my opinion, show a 'reconfigurable' platform.
- The final statement of the manuscript: "It should enable the observation of PT-symmetric beam dynamics, experimental work on non-Hermitian spectral band engineering with sub-wavelength periodicity, optical realization of encircling EPs with geometric phase effects, and attendant device applications such as switchable unidirectional couplers. The "Migration to integrated optics platforms is also envisaged" statement is not explicitly supported by the presented results and should be presented as speculation, or removed.
- In Figure 4a (and 1bc) I am unclear as to whether the points presented represent a single or averaged measurement. If it is the latter, error bars should be included and defined (going back to the repeatability point previously mentioned). In Figure 4c, the error bars should be defined in the figure caption.

- The manuscript is clearly-written throughout and concise. However, as many interested readers would be coming from a PT-symmetry/microfabrication background, additional discussion of the advantages/disadvantages of azo-polymer use would be beneficial.

- Finally, I call on the authors to verify the following equations in the Supplementary:

-2nd page: “... state parameter $\alpha = \tan^{-1}\left(\frac{|E_{py}|}{|E_{px}|}\right)^2$...”

Please verify whether you mean $\alpha = \tan^{-1}\left(\frac{|E_{py}|}{|E_{px}|}\right)$ for this and the main text on the 3rd page.

-4th page. In Anti-symmetric mode, the authors wrote: “ $\alpha_1 \cong -\frac{Qk_0}{2\sqrt{\epsilon_{avg}}}$.” They can remove the approximation, because $\alpha_1 = -\frac{Qk_0}{2\sqrt{\epsilon_{avg}}}$.

-5th page. “ $C_{12} = \frac{||\eta_{+1}|^2 - |\eta_{-1}|^2|}{\sqrt{(|\eta_{+1}|^2 + |\eta_{-1}|^2)((|\eta_{+1}|^2 + |\eta_{-1}|^2) + (|\eta_{-1}\eta_{+1}|/Q)^2)}} \dots \dots$ ”

$C_{12} = \frac{||\eta_{+1}|^2 + |\eta_{-1}|^2 - 2\eta_{-1}\eta_{+1}|}{\sqrt{(|\eta_{+1}|^2 + |\eta_{-1}|^2 + Q^2)((|\eta_{+1}|^2 + |\eta_{-1}|^2) + (|\eta_{-1}\eta_{+1}|/Q)^2)}}$ ”

Consider:

$$c_{12} = \frac{||\eta_{+1}|^2 - |\eta_{-1}|^2|}{\sqrt{(|\eta_{+1}|^2 + |\eta_{-1}|^2)((|\eta_{+1}|^2 + |\eta_{-1}|^2) + (4|\eta_{-1}\eta_{+1}|/Q)^2)}}$$

$$c_{12} = \frac{||\eta_{+1}|^2 + |\eta_{-1}|^2 - 2\eta_{-1}\eta_{+1}|}{\sqrt{(|\eta_{+1}|^2 + |\eta_{-1}|^2 + Q^2)((|\eta_{+1}|^2 + |\eta_{-1}|^2) + (4|\eta_{-1}\eta_{+1}|/Q)^2)}}$$

-6th page. Eq. S23: “ $\mathbf{H} = \frac{k_0}{2\sqrt{\epsilon_{avg}}} \begin{bmatrix} -\frac{1}{4}\left(\frac{K}{2k_0}\right)^2 & \eta_{-1} \\ \eta_{+1} & -\frac{1}{4}\left(\frac{K}{k_0}\right)^2 \end{bmatrix}$ ”

Consider:

$$\mathbf{H} = \frac{k_0}{2\sqrt{\varepsilon_{avg}}} \begin{bmatrix} -\frac{1}{4}\left(\frac{K}{k_0}\right)^2 & \eta_{-1} \\ \eta_{+1} & -\frac{1}{4}\left(\frac{K}{k_0}\right)^2 \end{bmatrix}$$

-6th page. Eq. S25: “ $A_1(d) = E_1 e^{j\Phi} + i\eta_{+1}\rho E_2$; $A_2(d) = E_2 + i\eta_{-1}\rho E_1 e^{j\Phi}$ ”

Consider:

$$A_1(d) = E_1 e^{j\Phi} + i\eta_{-1}\rho E_2$$

$$A_2(d) = E_2 + i\eta_{+1}\rho E_1 e^{j\Phi}$$

-6th page. Eq. S26: “

$$I_1(d) = |A_1(d)|^2 = |E_1|^2 + \rho^2 |\eta_{+1} E_2|^2 + 2E_1 E_2 \rho \operatorname{Re}\{-i\eta_{+1}^* e^{j\Phi}\}$$

$$I_2(d) = |A_2(d)|^2 = |E_2|^2 + \rho^2 |\eta_{-1} E_1|^2 + 2E_1 E_2 \rho \operatorname{Re}\{i\eta_{-1} e^{j\Phi}\}$$

“

Consider:

$$I_1(d) = |A_1(d)|^2 = |E_1|^2 + \rho^2 |\eta_{+1} E_2|^2 + 2E_1 E_2 \rho \operatorname{Re}\{-i\eta_{-1}^* e^{j\Phi}\}$$

$$I_2(d) = |A_2(d)|^2 = |E_2|^2 + \rho^2 |\eta_{-1} E_1|^2 + 2E_1 E_2 \rho \operatorname{Re}\{i\eta_{+1} e^{j\Phi}\}$$

-6th page. Eq. S27: “

$$I_1^{INT} - I_2^{INT} = 2E_1 E_2 \rho \operatorname{Im}\{\eta_{+1}^* + \eta_{-1}\}$$

$$I_1^{INT} + I_2^{INT} = 2E_1 E_2 \rho \operatorname{Im}\{\eta_{+1}^* - \eta_{-1}\}$$

“

Consider:

$$I_1^{INT} - I_2^{INT} = 2E_1 E_2 \rho \operatorname{Im}\{\eta_{-1}^* + \eta_{+1}\}$$

$$I_1^{INT} + I_2^{INT} = 2E_1 E_2 \rho \operatorname{Im}\{\eta_{-1}^* - \eta_{+1}\}$$

In general, I invite the authors to revise their manuscript to address the above specific concerns, before a final decision is reached for publication in Nature Communications.

Reviewer #3 (Remarks to the Author):

In this manuscript, the authors experimentally generate a non-hermitian photonic lattices and later use them to experimentally study wave propagation in such lattices. In particular they report the precise experimental observation of an exceptional point --- a phase transition from real to complex spectra.

I find the manuscript original and report on an important experimental results which will have impact on the field of PT photonics and optics.

The authors need to add the following references:

1. J. Doppler et.al. "Dynamically encircling exceptional points in a waveguide: asymmetric mode switching from the breakdown of adiabaticity",
<http://arxiv.org/ftp/arxiv/papers/1603/1603.02325.pdf>

In that paper, the authors also report on the experimental observation of dynamically encircling exceptional points in a waveguide. I ask the current authors to remark and address the question, how their results compares and contrast with those obtained by Doppler et.al.

2. When the authors mention unconventional beam dynamics in the abstract, they need to add the following three references:

K. G. Makris, Ziad H. Musslimani, D. N. Christodoulides and Stefan Rotter, Constant intensity waves and their modulation instability in non-hermitian potentials, *Nature Communications*, 6, 7257 (2015). In this paper, constant intensity waves can propagates in non-homogeneous PT symmetric media without any scattering -- this is highly counter intuitive phenomena that is being attributed to the presence of gain and loss.

The other paper is

Mark J. Ablowitz and Ziad H. Musslimani, Integrable nonlocal nonlinear Schrödinger equation, *Physical Review Letters*, 110, 064105 (2013). In this paper, the authors present the first PT symmetric nonlinear Schrodinger like equation which is completely integrable.

Also, the authors need to reference:

M. Wimmer, et.al. Observation of optical solitons in PT-symmetric lattices, *Nature Communication*, 6, 7782 (2015).

I am willing to accept the manuscript once the authors make the above changes.

Author response to the review report

Manuscript # NCOMMS-16-02610-T

Manuscript title: **Observation of exceptional points in reconfigurable non-Hermitian vector-field holographic lattices**

Authors: Choloong Hahn, Youngsun Choi, Jae Woong Yoon, Seok Ho Song, Cha Hwan Oh, and Pierre Berini.

We sincerely appreciate the reviewers for their valuable comments and for their time taken to provide the substantive reviews. In response to the review report, we have revised our manuscript as elaborated below.

Reviewer #1 (Remarks to the Author):

Hahn and co-workers report a nice approach to construct complex gratings using a clever vector holographic interference illumination pattern of an azobenzene-doped polymer thin film. The authors provide a firm theoretical description and their experiments clearly demonstrate strong asymmetric diffraction between the +1/-1 orders at the exceptional point. The basic approach is flexible and adds to a relatively small handful of experimental routes to realize complex optical potentials, therefore I believe it will be a valuable contribution to the non-Hermitian optics community. A few points that would be helpful to address:

Reviewer Comment 1-1. There is an existing report by Birabassov et. al., [Opt. Lett. 24, 1669 (1999)] that also describes asymmetric +1/-1 order diffraction through construction of a complex grating in an organic thin film through interference pattern illumination. The approach is different than that employed here, however, I would have expected it to be cited, particularly since Birabassov also used the azo dye Disperse Red 1.

Author Response 1-1) We have added this reference paper with an additional sentence “It is also higher than the reported value of $\Gamma \sim 7\pm 1$ generated by a complex photonic lattice in an azo-polymer film by Birabassov *et al.*²⁷ Their approach is based on spatially-modulated spectral hole burning that permanently changes the material properties in an irreversible manner.” in the third line from bottom on page 6.

Reviewer Comment 1-2. A discussion of what it takes to write a temporally-stable complex grating would be helpful. While not an issue for the present experiments, in general, having a system that is continually changing in the manner of Fig. 4a is likely to hamper broader experimental use of this approach for exploring non-Hermitian optical potentials as well as some device applications. Additionally, providing a sense of how long it takes for the gratings to dissipate once the write beams are turned off would be helpful (at a given sample temperature).

Author Response 1-2) In order to provide descriptions on these issues of temporally stable lattice generation and turning-off responses, we have added a new paragraph “Although the results summarized in Fig. 4 are taken from temporally-varying complex lattices, stationary lattices with $\xi = \xi_\infty$ are obtained when the pump fields are continuously applied over a recording time $t \gg \tau_R$ which is of the order of 10 min in our experimental configuration. Here, we have selected a sufficiently low ξ_∞ (≈ 0.28) to produce a temporally-tuned parametric scan over a sufficiently broad parameter range to clearly observe exceptional point behavior. The ξ_∞ value is in principle adjustable to higher values (up to 0.5) by tuning the pump polarization parameters. Alternatively, incorporating a dielectric layer on top of an azo-polymer film provides another way to tune the ξ_∞ value, following the elastic modulus and thickness of the dielectric layer. Turning off the pump fields leads to complete annihilation of the imaginary sub-lattice in a few seconds due to thermal relaxation at room temperature as the photo-isomerization and consequent molecular reorientation processes cease to occur. In contrast, the generated real sub-lattice persists in the form of a surface-relief grating until another pump or erasing fields drive new periodic gradient forces or planarizing potentials, respectively.” at the beginning of **Discussion** section on page 7.

Reviewer Comment 1-3. The authors discuss a number of possible extensions of this approach toward the end of the manuscript, but notably do not mention the possibility of incorporating gain. As gain is an important aspect for broader investigations and applications of PT symmetry, I would expect at least some mention of routes to accomplish this (e.g. addition of a laser dye into the polymer film) and whether there are any particular challenges envisioned.

Author Response 1-3) For this issue, we have revised the last part of the conclusion with new sentences “For instance, introducing optical gain by doping azo-polymer films with laser dyes not only compensates average losses but also may enable observation of much broader classes of non-Hermitian optical effects including saturated-gain-induced nonreciprocal light transmission,^{35,36} spectral singularities and associated unconventional laser oscillation effects,⁶ optical solitons,^{11,13} and uniform-intensity Bloch wave beams.¹⁵ On this basis, we envisage further experimental work on non-Hermitian spectral band engineering with sub-wavelength periodicity, the optical realization of encircling EPs with

Reviewer #2

RECOMMENDATION

Invite the authors to revise their manuscript to address specific concerns before a final decision is reached

COMMENTS TO THE AUTHOR(S)

The manuscript by Hahn et al. entitled “Observation of exceptional points in reconfigurable non-Hermitian vector-field holographic lattices” presents experimental results of an azobenzene-doped polymer thin film configured to exhibit a PT-symmetric Hamiltonian. The results show good agreement with numerical models (also presented) and the authors claim the first experimental demonstration of reconfigurable non-Hermitian photonic lattices in the optical domain and observe the purest exceptional point to date. The findings enable a versatile platform for PT-symmetry investigations and device studies.

The work is, in the opinion of this reviewer, quite nice although some points raised in the manuscript should be expanded upon and clarified. The following comments substantiate this reviewer’s opinion that the manuscript should be revised before a decision is reached:

Reviewer Comment 2-1. General interest: The work deals with the investigation of a new platform for studies of optical PT-symmetry, a very 'hot' field of research, with many interesting phenomena and applications already demonstrated within the last decade. I believe this context is well presented in the paper. While I believe the manuscript will be of interest to others in the field due to its main claim of a reconfigurable non-Hermitian photonic lattice (note that this platform's accessibility versus competing platforms should be further discussed), I feel this claim could be made more convincing. Only in the concluding paragraph is it referenced that holograms in azo-polymers are rewritable (Ref. 28) and if the reconfigurability is to be the main claim, it should be more apparent when multiple configurations using the same azo-polymer are used (aka this part should be rewritten). The repeatability and ability to show different non-Hermitian configurations with a single film must be demonstrated to, in my opinion, show a 'reconfigurable' platform.

Author Response 2-1) To clearly explain reconfigurability of the complex photonic lattices due to our approach, we have added a following new paragraph at the end of **Discussion** section on page 7~8:

“Previously, holographic lattice generation in azo-polymers has been widely studied to realize optical memory devices permitting stable reconstruction of data with multiple read-write-erase cycles below a certain photo-bleaching threshold intensity.³³ In our case, we have confirmed that significant photo-bleaching is observed at a pump intensity of 1.7 W cm^{-2} for a recording time of over 2 hours. In Supplementary Note 4, we provide an experimental result on the complex lattice reconfiguration that show stable formation of reconfigured complex lattices on a single sample spot with multiple write-erase-rewrite cycles. We confirm a stable reconfiguration time of $\sim 1 \text{ min}$ for a pump intensity of 1.7 W cm^{-2} for a total recording time of 40 min and a robust contrast ratio tuning in response to the varying pump-polarization parameters. In the experiment that leads to the data presented in Fig. 4, we have used again a single sample spot for multiple write-erase-rewrite cycles at a lower pump intensity of 30 mW cm^{-2} . In this case, no photo-bleaching effects were identified and reconfiguring a complex lattice into another form takes a fairly longer time - about 10 min, as required for the stable formation of the real sub-lattice. In fact, a much shorter reconfiguration time is possible with a slight modification of the sample geometry. We note that creation and annihilation of the imaginary sub-lattice takes a few seconds, following the relaxation time of azo-dyes into the stable trans-phase state aligned in a preferred orientation for a given pump field condition. Therefore, if a real sub-lattice is predefined, *e.g.*, using a standard lithographic technique or other available means, one can readily obtain stable reconfiguration within a few seconds using this strategy.”

Supplementary Note 5 has also been added in association with this revision point. Experimental data and associated explanations for the complex-lattice reconfiguration with multiple write-erase-rewrite cycles on a single sample spot are provided therein as appended below:

Supplementary Note 4: Reconfiguration of complex lattices with multiple write-erase-rewrite cycles

Establishing the reconfigurability of complex photonic lattices due to our proposed method, we perform a time-domain measurement of diffraction efficiencies $I_{\pm 1}$ and contrast ratio under multiple write-erase-rewrite cycles on a single sample spot. The results are shown in Fig. S5. In this experiment, pump intensity is fixed at 1.7 W cm^{-2} and the pump-polarization parameters are given in the caption. Figure S5(a) shows temporal profiles of pump intensities for the two pump beams. Strong transient noise in the pump intensity for the initial time range $< 200 \text{ s}$ is due to fluctuation of the Nd:YAG laser during stabilization. We set the pump-1 intensity to be switched on/off with a regular shutter with sub-ms switching speed while the pump-2 intensity remains constant. Slight modulation of the pump-2 intensity coincident with the pump-1 switching is due to a weak diffraction of pump 1 toward the photodetector monitoring pump 2. This configuration is

intended to make the pump-2 beam erase surface-relief grating (real sub-lattice) component during pump 1 is turned off. In addition, different ψ_1 values are applied for each pump-1 on/off cycle. Therefore, the series of the pump-1 on/off cycles produces multiple write-erase-rewrite (WER) cycles for different complex lattice configurations on a single azo-polymer sample spot. Measured $I_{\pm 1}$ and corresponding contrast ratio profiles are shown in Figs. S5(b) and S5(c). Although period of WER cycles are uneven in a range of 50 s ~ 160 s since we controlled ψ_1 rotation and shutter on/off status manually, it is clearly confirmed that the diffraction intensities are stabilized within 40 s corresponding to the time for the surface-relief grating formation with pump intensity of 1.7 W cm^{-2} . Importantly, the contrast ratio profile in Fig. S5(c) shows a bell-shaped envelope in response to the change in ψ_1 . This confirms that the generated complex lattice configuration on a single sample spot changes with the pump polarization parameters.

Supplementary Figure S5 | Reconfiguration of a complex lattice under multiple write-erase-rewrite cycles at a single sample spot. (a) Pump intensity profiles. (b) Diffraction intensity $I_{\pm 1}$ profiles. (c) Corresponding contrast ratio I_{+1}/I_{-1} profile. In this measurement, ψ_1 is tuned from 0° to 133° as indicated in (a) for each writing cycle. Other pump polarization parameters are fixed at $e_1 = 1$, $e_2 = 0.67$, and $\psi_2 = 72.4^\circ$. The pump intensity is also fixed at 1.7 W cm^{-2} .

Reviewer Comment 2-2. The final statement of the manuscript: "It should enable the observation of PT-symmetric beam dynamics, experimental work on non-Hermitian spectral band engineering with subwavelength periodicity, optical realization of encircling EPs with geometric phase effects, and attendant device applications such as switchable unidirectional couplers. The "Migration to integrated optics platforms is also envisaged" statement is not explicitly supported by the presented results and should be presented as speculation, or removed.

Author Response 2-2) We accept the fact that the statement "*Migration to integrated optics platforms is also envisaged*" does not deliver a substantive information and thereby can be speculative. Consequently, we have removed this sentence from the conclusion. In addition, the revision due to **Author Response 1-3** further improves our conclusion in association with this comment.

Reviewer Comment 2-3. In Figure 4a (and 1bc) I am unclear as to whether the points presented represent a single or averaged measurement. If it is the latter, error bars should be included and defined (going back to the repeatability point previously mentioned). In Figure 4c, the error bars should be defined in the figure caption.

Author Response 2-3) In Figure 4 caption, we have added a statement of error bar definition as "The error bars correspond to the standard deviation due to the dark current of the photodetector which impacts Γ for small I_{-1} levels." With this statement, it is immediately understood that each curve of symbols in Fig. 4a is taken from a single measurement, not averaged over repeated measurements.

Reviewer Comment 2-4. The manuscript is clearly-written throughout and concise. However, as many interested readers would be coming from a PT-symmetry/microfabrication background, additional discussion of the advantages / disadvantages of azo-polymer use would be beneficial.

Author Response 2-4) The two added paragraphs due to **Author Responses 1-2** and **2-1** provides clear explanations on chances and limitations of our approach. Therefore, we do not have any further changes on this issue.

Reviewer Comment 2-5. Finally, I call on the authors to verify the following equations in the Supplementary:

(2-5a)

-2nd page: "... state parameter $\alpha = \tan^{-1}\left(\frac{E_{py}}{E_{px}}\right)$..."

Please verify whether you mean $\alpha = \tan^{-1}\left(\frac{E_{py}}{E_{nx}}\right)$ for this and the main text on the 3rd page.

(2-5b)

-4th page. In Anti-symmetric mode, the authors wrote: " $\alpha_1 \cong -\frac{Qk_0}{2\sqrt{\varepsilon_{avg}}}$." They can remove the approximation, because $\alpha_1 = -\frac{Qk_0}{2\sqrt{\varepsilon_{avg}}}$.

(2-5c)

-5th page. " $c_{12} = \frac{||\eta_{+1}|^2 - |\eta_{-1}|^2|}{\sqrt{(|\eta_{+1}|^2 + |\eta_{-1}|^2)((|\eta_{+1}|^2 + |\eta_{-1}|^2) + (|\eta_{-1}\eta_{+1}|/Q)^2)}} \dots \dots$ "

$$c_{12} = \frac{||\eta_{+1}|^2 + |\eta_{-1}|^2 - 2\eta_{-1}\eta_{+1}|}{\sqrt{(|\eta_{+1}|^2 + |\eta_{-1}|^2 + Q^2)((|\eta_{+1}|^2 + |\eta_{-1}|^2) + (|\eta_{-1}\eta_{+1}|/Q)^2)}}$$

Consider:

$$c_{12} = \frac{||\eta_{+1}|^2 - |\eta_{-1}|^2|}{\sqrt{(|\eta_{+1}|^2 + |\eta_{-1}|^2)((|\eta_{+1}|^2 + |\eta_{-1}|^2) + (4|\eta_{-1}\eta_{+1}|/Q)^2)}}$$

$$c_{12} = \frac{||\eta_{+1}|^2 + |\eta_{-1}|^2 - 2\eta_{-1}\eta_{+1}|}{\sqrt{(|\eta_{+1}|^2 + |\eta_{-1}|^2 + Q^2)((|\eta_{+1}|^2 + |\eta_{-1}|^2) + (4|\eta_{-1}\eta_{+1}|/Q)^2)}}$$

(2-5d)

-6th page. Eq. S23: " $\mathbf{H} = \frac{k_0}{2\sqrt{\varepsilon_{avg}}} \begin{bmatrix} -\frac{1}{4} \left(\frac{K}{2k_0}\right)^2 & \eta_{-1} \\ \eta_{+1} & -\frac{1}{4} \left(\frac{K}{k_0}\right)^2 \end{bmatrix}$ "

Consider:

$$\mathbf{H} = \frac{k_0}{2\sqrt{\varepsilon_{avg}}} \begin{bmatrix} -\frac{1}{4} \left(\frac{K}{k_0}\right)^2 & \eta_{-1} \\ \eta_{+1} & -\frac{1}{4} \left(\frac{K}{k_0}\right)^2 \end{bmatrix}$$

(2-5e)

-6th page. Eq. S25: " $A_1(d) = E_1 e^{j\Phi} + i\eta_{+}\rho E_2$; $A_2(d) = E_2 + i\eta_{-}\rho E_1 e^{j\Phi}$ "

Consider:

$$A_1(d) = E_1 e^{j\Phi} + i\eta_{-1}\rho E_2$$

$$A_2(d) = E_2 + i\eta_{+1}\rho E_1 e^{j\Phi}$$

(2-5f)

-6th page. Eq. S26: "

$$I_1(d) = |A_1(d)|^2 = |E_1|^2 + \rho^2 |\eta_{+1} E_2|^2 + 2E_1 E_2 \rho \text{Re}\{-i\eta_{+1}^* e^{j\Phi}\}$$

$$I_2(d) = |A_2(d)|^2 = |E_2|^2 + \rho^2 |\eta_{-1} E_1|^2 + 2E_1 E_2 \rho \text{Re}\{i\eta_{-1} e^{j\Phi}\}$$

Consider:

$$I_1(d) = |A_1(d)|^2 = |E_1|^2 + \rho^2 |\eta_{+1} E_2|^2 + 2E_1 E_2 \rho \text{Re}\{-i\eta_{-1}^* e^{j\Phi}\}$$

$$I_2(d) = |A_2(d)|^2 = |E_2|^2 + \rho^2 |\eta_{-1} E_1|^2 + 2E_1 E_2 \rho \text{Re}\{i\eta_{+1} e^{j\Phi}\}$$

(2-5g)

-6th page. Eq. S27: "

$$I_1^{INT} - I_2^{INT} = 2E_1 E_2 \rho \text{Im}\{\eta_{+1}^* + \eta_{-1}\}$$

$$I_1^{INT} + I_2^{INT} = 2E_1 E_2 \rho \text{Im}\{\eta_{+1}^* - \eta_{-1}\}$$

Consider:

$$I_1^{INT} - I_2^{INT} = 2E_1 E_2 \rho \text{Im}\{\eta_{-1}^* + \eta_{+1}\}$$

$$I_1^{INT} + I_2^{INT} = 2E_1 E_2 \rho \text{Im}\{\eta_{-1}^* - \eta_{+1}\}$$

Author Response 2-5) We greatly appreciate Reviewer #2 for this comment and for taking his valuable time to review our formulation. These errors have been corrected accordingly. In particular, the suggested constant factor "4" for c_{12} and c_{23} as a correction in item (2-5c) should be "2". All other indications of errors in items (2-5a,b,d~g) by Reviewer #2 are correct. In addition, we have confirmed after multiple rounds of careful reviews on our own that all the results included in our original manuscript are not affected by these typographical errors.

In general, I invite the authors to revise their manuscript to address the above specific concerns, before a final decision is reached for publication in Nature Communications.

Reviewer #3:

In this manuscript, the authors experimentally generate a non-hermitian photonic lattices and later use them to experimentally study wave propagation in such lattices. In particular they report the precise experimental observation of an exceptional point --- a phase transition from real to complex spectra.

I find the manuscript original and report on an important experimental results which will have impact on the field of PT photonics and optics.

The authors need to add the following references:

Reviewer Comment 3-1. J. Doppler et.al. "Dynamically encircling exceptional points in a waveguide: asymmetric mode switching from the breakdown of adiabaticity", <http://arxiv.org/ftp/arxiv/papers/1603/1603.02325.pdf>

In that paper, the authors also report on the experimental observation of dynamically encircling exceptional points in a waveguide. I ask the current authors to remark and address the question, how their results compares and contrast with those obtained by Doppler et.al.

Author Response 3-1) We have added this reference paper as [31] with an additional description "We note that reconfigurable non-Hermitian systems are of great importance for exploring chiral EP dynamics^{28,29} and associated asymmetric state interchange between orthogonal eigenmodes.^{30,31} Although effects associated with EPs are observable in temporally-static, space-variant systems,³¹ experimental realization of reconfigurable non-Hermitian systems provides rigorous means to study true time-domain dynamics and tunable devices taking advantage of temporally-varying system parameters." at the end of second paragraph of **Discussion** section on page 7.

Reviewer Comment 3-2. When the authors mention unconventional beam dynamics in the abstract, they need to add the following three references:

K. G. Makris, Ziad H. Musslimani, D. N. Christodoulides and Stefan Rotter, Constant intensity waves and their modulation instability in non-hermitian potentials, Nature Communications, 6, 7257 (2015). In this paper, constant intensity waves can propagates in non-homogeneous PT symmetric media without any scattering -- this is highly counter intuitive phenomena that is being attributed to the presence of gain and loss.

The other paper is

Mark J. Ablowitz and Ziad H. Musslimani, Integrable nonlocal nonlinear Schrödinger equation, Physical Review Letters, 110, 064105 (2013). In this paper, the authors present the first PT symmetric nonlinear Schrodinger like equation which is completely integrable.

Also, the authors need to reference:

M. Wimmer, et.al. Observation of optical solitons in PT-symmetric lattices, Nature Communication, 6, 7782 (2015).

Author Response 3-2) These reference papers have been added as [12], [13], and [15]. In addition, we inform that the associated sentence has been moved from abstract to the second introductory paragraph as "Moreover, purely spatial, complex photonic lattices have showed unconventional beam dynamics associated with spectrally singular Bragg scattering,⁶ asymmetric or solitary optical propagation,¹¹⁻¹⁴ and counter-intuitive uniform-intensity wave solutions in non-uniform media.¹⁵" on page 2. This relocation is due to the format standard of the journal.

I am willing to accept the manuscript once the authors make the above changes.

In the revised manuscript, all changes are highlighted in yellow and indications of their positions are based on the revised manuscript. We additionally inform that we also have editorial changes to comply with the journal's format requirement as guided by MANUSCRIPT CHECKLIST at

http://www.nature.com/ncomms/authors/ncomms_manuscript_checklist.pdf.

Thank you very much for your consideration.

Apr. 18, 2016

Jae Woong Yoon, Seok Ho Song, Pierre Berini, Choloong Hahn, Youngsun Choi, and Cha Hwan Oh.

REVIEWERS' COMMENTS:

Reviewer #1 (Remarks to the Author):

The authors have satisfactorily addressed my original concerns.

Reviewer #2 (Remarks to the Author):

The authors have added additional discussion regarding the 'reconfigurability' of the platform, including additional measurements and details that significantly add to the quality of the manuscript. Furthermore, they clarified on the context and implications of the research (while also addressing Reviewer 1's concerns). Finally, they fixed the typographical errors I noted and assure these did not influence their manuscript results.

As I have no further concerns and the ones I did have were addressed in full, and as the manuscript presents work of high interest to the PT community (as discussed in my last response), I recommend the manuscript for publication.